# Costing of Cesarean Sections in a Government and a Non-Governmental Hospital in Cambodia—A Prerequisite for Efficient and Fair Comprehensive Obstetric Care

**DOI:** 10.3390/ijerph17218085

**Published:** 2020-11-02

**Authors:** Eva Glaeser, Bart Jacobs, Bernd Appelt, Elias Engelking, Ir Por, Kunthea Yem, Steffen Flessa

**Affiliations:** 1Department of General Business Administration and Health Care Management, University of Greifswald, 17489 Greifswald, Germany; eva.glaeser@posteo.de; 2Social Health Protection Programme, Deutsche Gesellschaft für Internationale Zusammenarbeit (GiZ), Phnom Penh 12302, Cambodia; bart.jacobs@giz.de (B.J.); bernd.appelt@giz.de (B.A.); elias.engelking@giz.de (E.E.); 3National Institute of Public Health (NIPH), Phnom Penh 12150, Cambodia; ipor@niph.org.kh (I.P.); kunthea.yem@niph.org.kh (K.Y.)

**Keywords:** cesarean section, costing, Cambodia, maternal health care, Asia

## Abstract

Knowing the cost of health care services is a prerequisite for evidence-based management and decision making. However, only limited costing data is available in many low- and middle-income countries. With a substantially increasing number of facility-based births in Cambodia, costing data for efficient and fair resource allocation is required. This paper evaluates the costs for cesarean section (CS) at a public and a Non-Governmental (NGO) hospital in Cambodia in the year 2018. We performed a full and a marginal cost analysis, i.e., we developed a cost function and calculated the respective unit costs from the provider’s perspective. We distinguished fixed, step-fixed, and variable costs and followed an activity-based costing approach. The processes were determined by personal observation of CS-patients and all procedures; the resource consumption was calculated based on the existing accounting documentation, observations, and time-studies. Afterwards, we did a comparative analysis between the two hospitals and performed a sensitivity analysis, i.e., parameters were changed to cater for uncertainty. The public hospital performed 54 monthly CS with an average length of stay (ALOS) of 7.4 days, compared to 18 monthly CS with an ALOS of 3.4 days at the NGO hospital. Staff members at the NGO hospital invest more time per patient. The cost per CS at the current patient numbers is US$470.03 at the public and US$683.23 at the NGO hospital. However, the unit cost at the NGO hospital would be less than at the public hospital if the patient numbers were the same. The study provides detailed costing data to inform decisionmakers and can be seen as a steppingstone for further costing exercises.

## 1. Introduction

The Cambodian health care system has improved a lot in the past decades, both in terms of quantity and quality of provided services. The country had one of the highest maternal mortality rates in South‑East Asia [1], whereby the strengthening of maternal and child health services was accorded special attention [2]. This encompassed ensuring availability of Comprehensive Obstetric Care, including enabling access to health care facilities offering Cesarean section (CS), a crucial intervention to guarantee safe delivery.

In addition to strengthening the services’ availability, much effort has been dedicated to improving their accessibility by decreasing associated financial burdens. These interventions include social health protection schemes like the Health Equity Fund for poor households and social health insurance for private formal workers and civil servants. Prior to this more systematic approach to social health protection, Cambodia experimented with a variety of health financing interventions like contracting, vouchers for maternal and child health services, pay for performance, or the midwifery incentive scheme.

With a current CS rate of 6.3% [3], Cambodia is still under the international recommended rate of 10–15%, suggesting unmet medical needs [4]. Institutional deliveries have increased tremendously in the past decade from 22% in 2005 to 83% in 2014 and are still rising [5], whereby an increase in the number of CS deliveries is foreseeable. Births by use of CS are more resource intensive than normal vaginal deliveries (NVD) due to a higher need of higher qualified personnel, materials, and drugs and a long hospital stay. Reliable information on the required resources and associated costs can help policymakers to plan and allocate an appropriate amount of resources.

Besides the public health care sector in Cambodia, there is a large and mainly unregulated private health care sector. Private health care providers can be for-profit and not-for-profit. The former ranges from quacks and informal drug sellers to high-tech private for-profit hospitals, while the latter are comprised of non-governmental organizations (NGOs) and national or international private philanthropic institutes [6]. Even though most services by private providers tend to be more expensive than those of public institutions, people frequently seek health care services at non-public providers [7]. Therefore, policymakers should include the private for-profit and NGO health care providers in their plans, especially when considering them to be contracted by social health protection schemes.

Costing of health care services is a prerequisite to provide accurate and adequate data to decisionmakers managing hospitals or the health care system. Decisions to equitably and efficiently allocate health care resources can only be made if reliable data are available [8]. For many low- and middle-income countries, however, costing information cannot be retrieved from routine systems and required data is not always available. Usually, the costing systems currently used at the hospital level cannot assign costs to specific health care services [9].

To date, there are no costing studies on CS available for Cambodia. A few studies assessed the cost of specific health care services or diseases (e.g., [10,11,12,13]). The latest published costing study assessed the unit costs of public health care services in several health centres and hospitals using a step down costing approach [13]. While it showed that costing was feasible with this methodology, the study can only inform about the costs of certain units like outpatient visits or inpatient days at certain departments. To get a comprehensive picture of the costs for specific services at a hospital, it is necessary to underpin the step down costing of the hospital with bottom up costing studies for commonly provided and resource intensive services [14]. Furthermore, the number of costing studies on CS in South (East) Asia is extremely low. While it is possible to calculate roughly the cost per bed day using a top down costing approach and estimate the cost per CS by multiplying this figure with the average lengths of stay [15], a precise costing will require a bottom up costing approach. To our knowledge, only one study from Bangladesh provides evidence on this issue with a sufficient degree of precision [16].

Therefore, this study aims to determine the cost of a CS using a bottom up costing approach at a high level of accuracy. The study was carried out in two hospitals: a public hospital that was also part of the aforementioned step-down costing study and an NGO-funded not-for-profit private hospital. Comparing the results of the two hospitals can reveal differences in cost structure and provide some initial insight into the relatively unexplored NGO health care sector in Cambodia.

## 2. Materials and Methods

### 2.1. State-of-the-Art

Costing of health care providers and specific interventions, such as CS, is standard in middle- and high-income countries [17]. Traditionally, costing follows a top down approach by allocating the cost of the entire institution to different cost centers (e.g., administration, labor ward, surgical ward). Afterwards, the respective costs of cost centers are allocated to costing units, e.g., one appendectomy or one CS. This approach is called step down costing [18,19]. Less frequently, costs are determined for each activity and accumulated over the entire process. Consequently, this approach is termed process or activity based costing [20]. For low-income countries, the number of studies focusing on specific conditions is limited. As Grimes et al. show in their systematic review, in particular, there are only very few studies on the cost of CS and none are on South-East Asia [21].

We follow an activity-based costing involving the following steps:(1)Determining the process by personal observation and analysis of medical documentation(2)Determining direct payments per activity on the process by analyzing accounting documentation (e.g., drugs, materials)(3)Determining time consumption per activity on the process by time-study (e.g., working time of nurse-midwife per CS)(4)Determining the indirect cost per CS by allocating overheads to the process

With this approach, we followed the international costing standard implemented in other studies [18].

### 2.2. Research Setting

The study was conducted at a public hospital and an NGO hospital in Cambodia. Table 1 shows some characteristics of the study hospitals. The hospitals were not selected randomly, but we decided to work in hospitals with which long term working experience existed. In this way, we were sure that we would be granted access to processes and data. The data collection took place from January to March 2019. Ethical clearance was given by the National Ethics Committee for Health Research (#015 NECHR).

The public hospital had 133 beds and 2 operating rooms with 188 employees in the year 2018. In general, the hospital works within a low resource setting, with most buildings dating from around 1950 and relatively limited medical and non-medical equipment. The hospital is funded by the government, which covers staff remuneration and several running expenditures and provides most drugs, consumables, and equipment. Other income is generated through user fees. In 2018, the hospital served 31,387 outpatients and 13,155 inpatients. Further, 4273 inpatients were admitted at the maternity ward, accounting for a total of 15,386 inpatient days.

The NGO hospital is relatively new and has a dedicated operating room for CS. In 2018, the hospital had 50 beds, of which 16 are located at the maternity ward, and had 32,219 outpatients and 3239 admissions. The maternity ward had 1054 patients, who account for 1921 inpatient days. In general, the buildings are quite new and well equipped. The hospital generates funds mainly by charging user fees which tend to be much higher than at the public hospital, although it operates an exemption system which gives poorer patients a discount of up to 100%. Other resources come from government donations (vaccines) and, to a large extent, from overseas.

### 2.3. Data Sample and Cost Categories

The study took the provider’s perspective and calculated the cost per CS based on the provider’s expenditures. The study aimed to calculate a cost function in relation to patient numbers and capacity utilization. For this purpose, all costs were classified into variable, fixed, and step fixed costs (see Table 2).

The variable costs include drugs and materials that were directly used for a CS patient when preparing a patient for a CS in the operating theatre and at the wards. It also included materials used for laboratory tests and echography, the sterilization unit, and the patient’s food. The consumption per CS patient was determined by direct observation, complemented with information collected from patient files and the ward’s order sheets. All consumed drugs and materials were costed using price lists provided by the hospital’s pharmacy and the warehouse.

Fixed costs include the cost of depreciation for buildings and equipment and staff overheads and other overhead costs. Linear depreciation was used with a lifespan of 25 years for buildings, 5 years for minor electronic equipment like computers, 15 years for mechanical equipment, and 10 years for all other equipment. Allocation of fixed costs to CS patients was calculated by multiplying the fixed costs with a factor reflecting the share of expenses for CS patients. Allocating factors used were the share of CS patients to the total number of surgeries in 2018, laboratory tests, imaging and sterilizations, inpatient days in the specific wards, and the hospital. Data were obtained from the hospital accountant.

Step fixed costs are a special case of fixed costs. They remain constant until a certain capacity level is reached, then they jump up on a higher cost level. This cost category includes the cost of medical personnel, like medical doctors, midwives, nurses, and anaesthetists. The gross salaries for each staff category were obtained from the hospital’s administration. Figure 1 illustrates the three different cost concepts. While fixed costs remain constant irrespective of the number of CS, variable costs correlate with the number of CS. Step fixed costs start as normal fixed cost. However, as soon as a certain threshold (T1) is reached, they jump to a higher level. Afterwards, they behave again like fixed costs until they reach another threshold (T2). Usually, the thresholds are the capacity limits, such as the working capacity of one staff member.

The unit costs included salaries, incentives, and overtime payments. Since staff members did not exclusively serve CS-patients, their share of monthly remuneration dedicated to CS patients was calculated by multiplying the costs per productive working hour with the hours needed to perform the maximal number of CS as per calculated capacities.

### 2.4. Allocating Time

The average time spent per CS patient was obtained by direct observation, whereby the observer followed patients from preparation for surgery until the completion of post-operative monitoring. This included 15 patients at the public hospital and 12 patients at the NGO hospital. Additionally, the observer spent one month at each hospital and observed the time spent on patient care after surgery.

Besides the tasks for direct patient care, midwives and nurses perform tasks at the ward that are not directly attributed to a specific patient. For instance, midwives and nurses need working time to organize their daily work, to hand-over responsibilities to the next shift, and to do general documentation of all patients. The time consumed by these tasks cannot be allocated to one individual patients. Consequently, the respective time was allotted to all patients according to patient-staff ratio as is common in activity-based costing [17].

The total time needed for a day of CS patient care was then calculated as the sum of the time for all process steps required for the daily care of a CS patient. The time spent per CS patient per day was then multiplied by the average length of stay (ALOS)—minus one day, which concerns the day of CS that was calculated separately—to indicate the time for all care given at the wards. As the day of discharge is not included in the ALOS reporting, the time spent on the ward by the CS patient during the discharge day was added to the ALOS to determine the total time for patient care.

### 2.5. Calculating the Influence of Frequency of CS

To calculate the cost of a single CS in relation to the total number of CS performed, it is required that one assesses the capacity (respective thresholds) of each staff category in addition to the marginal (additional) time spent to perform a CS. To calculate the capacity for CS, assumptions about the workload for other patients compared to CS patients had to be made. The workload for a normal vaginal delivery (NVD) was assumed to be equal to the workload for a CS for a midwife. The time requirement for a non-CS surgery was assumed equal for the doctors and the operating theatre (OT)-team at the public hospital, since the reported anaesthesia times for CS and other surgical interventions did not differ substantially. At the NGO hospital, it was distinguished between minor (>00:30 h), middle (00:30–03:00 h), and major surgeries (<03:00 h) to calculate the total occupied time. These differentiations were made based on the reported anaesthesia times taken from routine records.

The monthly workload for productive work was calculated for the current number of patients. The time for caretaking per CS patient following the surgical procedure was calculated by the ALOS for normal deliveries minus the observed time spent on caring for CS patients. Subtracting the monthly workload from the total working hours gave the free time that could be dedicated to additional patients. It was assumed that a maximal 85% of the monthly working time of a staff member could be productive for both hospitals.

The average unit costs for a CS at both hospitals were calculated as a function of the number of patients for CS. The following costing function was used:(1)$$C_{x_{t}}=\frac{C_{f}+\sum_{i\epsilon K}\left(1-trunc\left(\frac{x_{t}}{k_{i}}\right)\right)\cdot d_{i}}{x_{t}}+v,$$
with
*C_xt_*average unit cost per CS in period t in US$*C_f_*fixed costs per month in US$*d_i_*monthly costs of unit *i* in US$*x_t_*number of CS done in period t*k_i_*monthly capacity of unit *i* in US$*v*variable costs of one CS in US$*K*set of step fixed cost units

## 3. Results

### 3.1. Patients’ and Procedural Characteristics

To compare patient’s characteristics at both hospitals, CS-patient files from January 2019 were reviewed (see Table 3). At the NGO hospital, there were 39 CS patients, with one file missing compared to 86 CS patients at the public hospital. Of the latter, 72 files were retrieved.

At the NGO hospital, about one quarter underwent elective CS and the remainder had emergency CS. At the public hospital, all CS were marked as an emergency. The main reasons for CS at the NGO hospital were reported as foetal distress (32%), no progress (21%), and breech or transverse presentation (18%), while at the public hospital, the main reasons were cephalon-pelvic disproportion (25%), breech or transverse presentation (17%), or a previous CS (15%).

At the NGO hospital, 76% of CS patients were discharged after three days. Patients staying more than five days usually arrived several days before CS due to complications during pregnancy (e.g., pre-eclampsia). At the public hospital, ALOS was seven days, with 39% staying more than seven days and 19% more than nine days. The calculated ALOS was 3.4 days at the NGO hospital and 7.4 at the public hospital. For the latter hospital, patients stayed on average 2.0 days at the Intensive Care Unit (ICU) and 5.4 days at the Maternity Ward.

### 3.2. Costing

At the NGO hospital, patients spent 01:44:29 h (standard deviation (SD) ± 00:25:17 h) in the OT, while this was 00:54:09 h (SD ± 00:10:45 h) in the public hospital. There was no substantial difference in duration observed for elective and emergency CS.

The time spent for a CS and postpartum care by staff group is shown in Table 4. There were substantial differences in observed durations in time spent by the obstetrician on CS patients. This may be related to the number of patients at the ward with an average of five in the maternity ward of the NGO hospital vs. 42 in the public hospital. Different staff-patient ratios also existed for nursing staff: at the NGO hospital, this was 1:2.6 for a midwife vs 1:13.2 for an ICU-nurse and 1:8.5 for a midwife at the maternity ward in the public hospital.

The different staffing levels and time requirements for a CS patient result in the different staff’s capacity to perform CS (see Table 5).

With the observed workloads, a midwife at the NGO hospital can take care of 2.5 CS patients per month, while at the public hospital, the capacity is 4.4 CS per midwife and month.

At the NGO hospital, several obstetricians were on duty during the daytime and on-call during nights and weekends, while at the public hospital, there was always one obstetrician on duty for 24 h. The obstetrician at the public hospital spent limited time at the maternity ward and only tended to stay there if a patient had complications. At the NGO hospital, obstetricians visited all patients daily and spent a substantial amount of time at the maternity ward. Therefore, the capacity to perform CS was higher at the public than at the NGO hospital for an obstetrician.

For variable costs, the NGO hospital spent US$62.58 per CS patient, compared to US$236.05 at the public hospital (Table 6). At the NGO hospital, 29.6% of variable costs per CS patient were made up of drugs and 32.5% materials; at the public hospital, 56.4% was for drugs and 32.8% materials.

An overview of the fixed costs for both hospitals is given in Table 7. The category “other overhead costs” include the costs of utilities, domestic supplies, maintenance, training, travel, transportation, exemptions from user fees, and other expenditures. The total fixed costs are US$1,148,388.34 at the NGO hospital and US$505,678.16 at the public hospital. The actual share of the fixed cost for CS was calculated by multiplying the fixed costs with a factor reflecting the share of expenses for CS patients. The NGO hospital OT was solely used for CS (factor for CS patients 100%), while CS accounted for 45% of all surgeries at the public hospital (factor for CS patients 45%). As such, the share of monthly fixed cost for CS patients was US$6444.58 for the NGO hospital and US$8529.59 for the public hospital.

The average unit costs depend on the number of CS performed (Table 8, Figure 2). The average cost per CS is shown in Table 8. The costs per CS inversely correlated with the number of CS. The leaps in the curves are produced by the step fixed costs. When the maximum capacity of a unit is reached, then the step fixed costs jump to a higher level and increase the average costs per CS. With 18 monthly CS at the NGO hospital and 54 monthly CS at the public hospital, the unit costs per CS patient were US$683.23 and US$470.03, respectively. If both hospitals performed the same number of CS per month, the unit cost would be less at the NGO hospital.

### 3.3. Sensitivity Analysis

The calculation of the average CS costs is based on several parameters that are subject to uncertainty. To estimate the influence of changes in the input parameters, a sensitivity analysis was performed. Besides different inputs for variable and fixed costs, the assumptions for the calculation of the staff’s capacity was varied (time consumption for other deliveries, time consumption for other duties, unproductive time). The variation in the respective parameter is given in Table 9. All other parameters stay the same as in the basic scenario.

It is obvious that the unit costs increase with higher fixed or variable costs. Changes in the variable costs have a higher influence at the public hospital, where variable costs account for a larger share in total costs. If staff need more time for other patients or productive work, the changes in cost are relatively small. If the unproductive time increases the costs per CS rise by 5.4% at the NGO hospital and + 2.5% at the public hospital if the unproductive time of all staff is increased by 10%.

## 4. Discussion

These results must be seen in the context of the Cambodian health care system and the international literature. Within the Cambodian environment, this is the first study calculating the average unit cost of CS in an NGO and a public hospital in Cambodia. The actual unit cost of a CS at US$683.23 (NGO) or US$470.03 (public) seems low for a high-income country but is very high for a lower middle-income country like Cambodia. Performing a CS is resource intensive in the context of limited health care budgets. Consequently, further analysis to improve the efficiency of an essential health care service like CS is crucial to make the best use of available resources.

Currently, the CS rate in Cambodia is 6.3%, i.e., about every 16th birth will incur such a high cost. However, in comparison to other countries and WHO standards, this rate (and consequently, the health care cost) might be too low. It is estimated that a minimum proportion of CS in resource poor settings should be about 10–15% in order to provide a safe delivery, even for complicated cases [23,24,25]. This low rate may suggest that the poorest have insufficient financial and/or physical access to CS services [5,26]. A doubling of the CS rate by improving access will result in higher costs to the provider. However, this does not include a linear increase of costs as the majority of costs of CS are fixed costs. The two studied hospitals differed in the number of treated patients and the organization of processes. The NGO hospital was relatively new with a rather limited number of inpatients compared to the public hospital which served a fourfold the number of inpatients. CS patients at the public hospital stayed substantially longer following surgery and were admitted to the ICU, while at the NGO hospital, such patients were admitted only at the maternity ward.

Patients at the NGO hospital spent a substantially longer time at the OT than at the public hospital. Most of this difference in duration is caused by the surgical procedure itself, while the time for preparation and postprocessing is similar. Possibly, this difference results from the experience of the staff members: at the public hospital, the concerned staff members have been working at the hospital for a long time with an average of nearly two daily CS, while the NGO hospital employs many young obstetricians with a CS happening every other day.

At the NGO hospital, patients stayed at the maternity ward throughout hospitalization and were cared for by midwives using the concept of total patient care with all care for a single patient being given by a single midwife. The midwife monitored the patient’s condition several times a day with vital sign checkups. At the public hospital, in contrast, the workflow was organized as functional care, which involves less time per patient. While CS patients stayed twice as long at the public hospital than at the NGO hospital, the total caring time spent by midwives and nurses was still the highest at the NGO hospital. While determining the appropriate level of care and ALOS are beyond the scope of this article, the time study showed that there are large variations between the two assessed hospitals.

The NGO hospital had high costs for the depreciation of buildings and equipment. No accurate information about the cost of buildings was available, but the directors gave an estimation of the total cost of setting up all hospital buildings. By attributing these costs to the total surface area of hospital buildings, a price of US$500 per square meter was estimated. This price is substantially higher than the price of the public hospital where original building prices were available from records. Most buildings of the public hospital were built more than 60 years ago, which may explain the observed differences in price per m^2^. In addition, the equipment depreciation for CS was much higher at the NGO hospital. The overhead costs for staff were also higher at the NGO hospital due to a larger number of staff members working at the service departments (97 vs. 50 at the public hospital).

Although the fixed costs at the NGO hospital were higher than at the public hospital, the share of fixed cost out of the total cost for CS patients was rather low at the NGO hospital. At both hospitals, CS patients accounted for 10% of all inpatient days. In contrast, the share of CS patients out of the total number of patient days was only 2% at the NGO hospital and 8% at the public hospital. Since the allocation according to equivalent inpatient days was used for several items, the share of fixed costs for CS patients was lower at the NGO hospital than at the public hospital.

The cost per CS was lower at the NGO hospital than at the public hospital when performing the same number of monthly CS procedures, mainly because of the large variable costs at the public hospital, over 50% of the unit cost per CS at the current number of patients. At the NGO hospital, the share of variable costs would only be around 25% when having one monthly CS.

The unit costs depend on the variables of the model. The most important variable was the number of CS per hospital. Increasing this number goes in hand with a cost reduction per procedure. Consequently, a concentration of CS in a limited number of hospitals will not only result in a higher number of operations per performing hospital, but also in lower unit and total costs of CS for the entire health care system. However, this might also increase the distance between the place of residence of the pregnant women and the hospital providing CS services, so direct non-medical costs (transport) will grow and the likelihood that pregnant mothers reach the hospital too late will increase as well. Health policymakers will definitely have to distinguish between urban and rural settings. For example, in neighboring Vietnam, urban women were twice as likely to undergo CS than rural women [27]. For Cambodia, this ratio was 2.7 in 2014 [28]. In Kenya, distance to the health facility was a main factor delaying timely access to CS [29]. At the same time, some degree of competition between providers might also increase the quality of services. Thus, the concentration of CS on fewer hospitals demands a very thorough and system wide analysis.

As the sensitivity analysis shows, the thresholds of step fixed costs is another important determinant of the CS unit cost. If a staff member can accommodate more work, it might be possible to shift the threshold to the right, thus reducing the unit costs. However, thresholds are not carved in stone but depend on the motivation and qualification of staff, whereby unit costs depend on these variables as well.

Consequently, the strong differences between the costs per CS in these two hospitals are due to different factors, including funding possibilities (resulting in higher fixed costs), length of stay, drug schemes, utilization, and—maybe—technical efficiency of the institution. Any costing system that does not distinguish between fixed and variable costs and does not develop a cost function will not be able to determine these influence factors. Thus, a detailed bottom up costing is much more precise and permits the leadership of the institutions to base their decisions on evidence. Top down costing systems are much easier but there is a risk that fixed costs are proportionalised, resulting in misleading results.

This study provides evidence for the management of the hospitals, the operational districts (OD), and the Government of Cambodia. The hospital management can compare its own costs and cost structures and analyze the factors influencing the reasons for these differences. This process has already started. For instance, it was realized that a major cost driver of the cost per CS in the Governmental hospital is the consumption of antibiotics. In this hospital, all patients receive this drug preventively after a CS, while this is not done in the NGO hospital. Furthermore, the leadership of the NGO hospital has realized that the occupancy rate is the most crucial factor, i.e., they have started discussing how to increase the utilization. Without doubt, these leaders have always had a “feeling” that there might be issues that have to be addressed, but now they have evidence and can start working on it.

The leadership of the OD as well as the staff of the ministry of health can also learn from these findings. However, they would require an evidence base from more institutions in order to conclude what to change. Consequently, it was decided to expand the costing of maternity services including normal deliveries in health centers and primary hospitals. The respective analysis will be based on the methodology presented in this paper so that the results will be comparable.

When resources are scarce and health needs are high, information on costs of services helps policymakers to allocate resources so that the population can access required health services in an equitable way [30,31]. Such economic evaluation information facilitates evidence based policymaking [32], something that is well recognized in high income countries. Conversely, using economic evaluations for evidence based policymaking in low- and middle-income countries is still challenging due to various factors, including insufficient local capacity to generate the information, lack of relationships between researchers and policymakers, and untimely dissemination of or too technical information [32]. Recently, many Asian countries have invested in Health Technology Assessment to overcome some of these obstacles [31,32]. The current study complements efforts to establish a routine health services costing system in Cambodia to enable policymakers to set prices for active purchasing of health services and to aid subnational level health managers to optimize service delivery with available resources [15].

The number of costing studies from low or lower middle-income countries is limited—in particular, for South (East) Asia. Thus, a reflection on our findings in light of the relevant literature is not very fruitful. In the region, only one study from Bangladesh provides reliable figures [16]. Similarly to the results of the present study, it compares the costs per CS at a public, a private for-profit, and an NGO-hospital. It found that the costs of a CS were lowest at an NGO-hospital (US$91, all cost data inflated to December 2018 values) compared to the private for-profit (US$117) and the public hospital (US$162). Outside of the region, a study from Uganda found that a CS was more expensive at a private non-for profit hospital (US$84.88) than at a public hospital (US$78.03) [33]. Another study determined the cost per CS at three public Brazilian hospitals as US$606.21 [34]. However, comparing available costing data from different countries is difficult due to different medical standards and prices for goods or services. Furthermore, the available studies are outdated and do not reflect medical inflation rates.

Calculating the cost of a CS using available top down costing data from Cambodia also gives different results. The costs per inpatient day from the different top down costing studies were multiplied with the ALOS of CS-patients at the public hospital and inflated to 2018 values: data of Fabricant from 2001 suggest a cost per CS of US$90.84 [35], while this would be US$100.17 using Collins’ data from 2007 [36] and US$305.24 when data from 2016 by Jacobs et al. [15] are used. The increased costs over time likely reflect improved services, indicating the need to base policy decisions on up-to-date data and to repeat data collection regularly.

The higher costs found in this study may reflect a more detailed allocation of costs: while using top down costing, all patients of a certain ward are considered to incur the same costs, irrespectively of their condition, the bottom-up costing distinguishes between patients of the same ward with different diagnoses. Since CS-patients need a higher amount of drugs and materials and more care than an NVD patient, the costs per inpatient day of a CS are likely to be higher than the average costs per inpatient day at the maternity ward. Hence, the top down costing seems to be appropriate to give a first assessment of the costs for hospitals and their different departments but should be accompanied with more detailed bottom up studies, specifically for complex and resource consuming interventions.

### Limitations

This study faces a number of limitations. Obviously, we selected merely two hospitals. The selection was not random but was based on access to the institution and data. Thus, we selected hospitals with which we have had a working relationship for years. They might be “better” than other hospitals in Cambodia, resulting in lower unit costs than the average.

This study relies on primary data collection and information obtained from reports, order sheets, and invoices. Both sources have several limitations and respective data need to be handled with caution. The direct resource consumption in the form of drugs, materials, and staff time were determined via direct observations. Since the duration of the study was short and the observations were performed by one observer only, the sample is quite small. All observations were only done during the daytime, so estimations about the work done at night had to be made. Furthermore, only staff members involved in the direct care were observed, excluding staff members of service departments like laboratory and echography. All these limitations could be minimized by a larger team of observers who could do observations simultaneously over a longer period of time and across several departments.

Generally, observational studies pose the risk of observer bias. Additional sources of errors during the observations could be wrong identification of the observed processes or mistakes in the protocolling process.

The availability and quality of secondary data was suboptimal, especially at the public hospital. Some data was only available as hard copies, other was outdated. Access to the cashbook at the public hospital was not given, so data collected in 2016 for another study had to be taken. The problem of unreliable data is common for all costing studies and can only be overcome if the hospital management improves data collection and reporting.

Finally, there is a risk of comparing “apples and oranges”. For instance, the buildings of the public hospital are quite old and written off. Thus, the depreciation is rather small, resulting in lower unit costs. It would be necessary to calculate standard costs instead of actual costs for both institutions.

## 5. Conclusions

In the context of the impressive progress made in promoting institutional deliveries by, amongst others, providing geographic and financial access to obstetric services using several demand- and supply-side financing mechanisms, a further increase of CS can be expected. With about 375,000 annual deliveries [37] and a CS rate of 6.3% [3], an additional 15,000 CS per year will result if the target CS rate of 10% is to be reached [38]. This suggests cost-implications for the Cambodian health care system which are mainly funded through out-of-pocket payments by Cambodian households.

Cost reductions through efficiencies in public hospitals should, therefore, be carefully looked into. The comparison to an NGO hospital suggests potential cost reductions in several areas, with procurement practices and working practices of hospital staff being the most prominent. This would be greatly aided if management boards of public hospitals could be more proactively engaged in decision-making that reaches beyond translating and implementing guidelines and requirements from the national level to the hospital environment. The recent instigated decentralization process may aid this [39]. The current state of routine data systems, however, imposes challenges to act as sources of information to guide decision making. Bottom up costing studies can provide more detailed and accurate data and information for the respective hospitals in particular and public Cambodian hospitals in general, and enable the management to make evidence based decisions for more cost efficient business operations.

## Figures and Tables

**Figure 1 ijerph-17-08085-f001:**
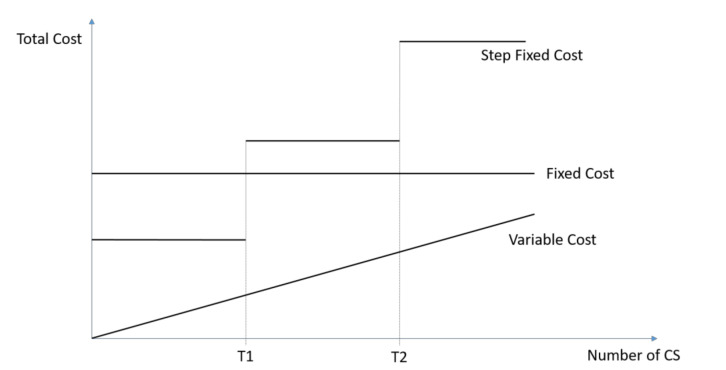
Variable, Fixed, and Step Fixed Cost.

**Figure 2 ijerph-17-08085-f002:**
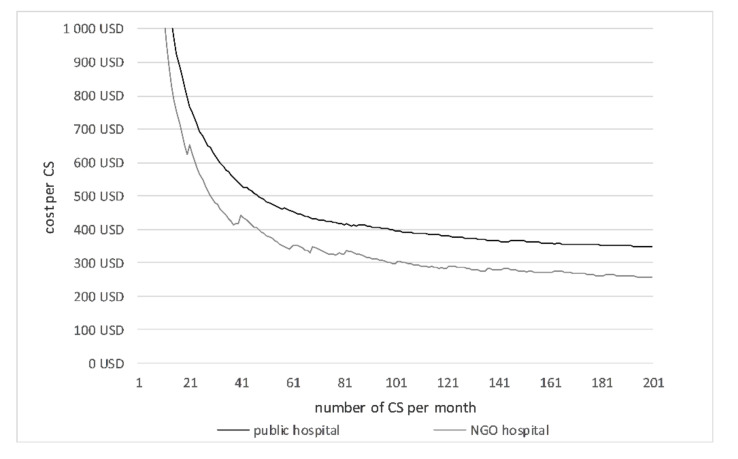
Average cost per CS.

**Table 1 ijerph-17-08085-t001:** Study hospitals (2018 data).

Key Variables	NGO Hospital	Public Hospital
Number of beds	60	133
Number of patients		
outpatient	32,219	31,387
inpatient	3293	13,155
Patients at the maternity ward	1054	4273
Patient days at the maternity ward	1921	15,386
Number of employees	221	188
Funding	User fees; overseas donations; government budget	Government budget; user fees

**Table 2 ijerph-17-08085-t002:** Costing terminology [22].

Costing Terminology	Definition	Example
Cost	Monetary value of resources consumed to produce a certain output	Monetary value of resources consumed to treat patients with Cesarean section (CS)
Variable cost	Costs that vary with output	Drugs and materials
Fixed cost	Costs that do not vary with output	Depreciation for buildings and equipment, overhead costs
Step fixed cost	A cost that does not change within certain high and low thresholds of activity, but which will change when these thresholds are breached	Cost of personnel
Total cost	Aggregation of all types of costs for all outputs in a certain period	Cost of treating all patients with CS
Average cost	Cost of all units divided by number of outputs	Average costs of treating one patient with CS
Marginal cost	Cost of one additional unit	Cost of treating one additional patient with CS
Unit cost	Cost per unit of output. Unit costs can refer to average unit cost and marginal unit cost	Cost of treating one patient with CS
Cost function	A formula used to predict the cost that will be experienced at a certain activity level	
Activity based costing	A costing method that divides the total production process into a number of activities and calculates the cost of each activity	
Cost center	Place where costs occur	Ward, theatre, administration, …
Service cost centers	Cost centers without direct contact to the product	Administration, kitchen, facility management

**Table 3 ijerph-17-08085-t003:** Caesarian section characteristics (in %).

Patients’ and Procedural Characteristics	NGO Hospital (*n* = 38)	Public Hospital (*n* = 72)
CS-rate	22.7	18.8
Type of CS		
elective	26	0
emergency	74	100
Reason for CS		
fetal distress/CTG non reassuring	32	8
no progress/failed induction	21	10
breech/transverse presentation	18	17
previous CS	11	15
cephalon-pelvic disproportion	3	25
other	18	24
ALOS (days)	3.4	7.4

CTG: Cardiotocography (used during pregnancy to monitor fetal heart rate and uterine contractions).

**Table 4 ijerph-17-08085-t004:** Average time spent per CS.

Time per CS	NGO Hospital	Public Hospital
Time of patient in OT	01:44:29 h	00:54:09 h
Time spent by doctors		
senior obstetricians	02:06:12 h	01:17:20 h
junior obstetricians	04:49:09 h	-
surgeons	-	01:09:31 h
anesthetists	02:03:35 h	
pediatrician	01:21:46 h	-
intern doctors	02:12:29 h	-
Time spent by other OT staff		
anesthesia nurses	02:22:43 h	02:07:27 h
scrub nurses	02:16:38 h	00:52:23 h
circulating nurses	02:19:11 h	02:08:16 h
Time spent by ward staff		
midwives	13:49:38 h	08:21:46 h
ICU nurses	-	02:15:03 h
psychologist	00:30:26 h	-

**Table 5 ijerph-17-08085-t005:** Monthly treatment capacities and unit costs per staff member.

	NGO Hospital		Public Hospital	
Cost Category	Monthly Capacity	Unit Cost Per Month (US$)	Monthly Capacity	Unit Cost Per Month (US$)
Senior obstetrician	20.3	443.65	28.9	285.00
Junior obstetrician	15.6		-	
Surgeon	-		72.7	
Anesthetist	68.3		-	
Pediatrician	10.2		-	
Intern doctor	14.2		-	
Anesthesia nurse	38.1	230.47	84.15	218.31
Scrub nurse	40.3		109.4	
Circulating nurse	39.4		83.0	
Midwife	2.5		4.4	
ICU-nurse	-		21.2	
Psychologist	28.1		-	

The unit cost averages for the group doctors and other medical staff are not given due to confidentiality. For the calculation of the costing function, the single unit costs per category were used.

**Table 6 ijerph-17-08085-t006:** Variable Cost (in US$).

Total Variable Cost Per CS	NGO Hospital US$ (%)	Public Hospital US$ (%)
Drugs	18.53 (29.6)	133.14 (56.4)
Materials	20.35 (32.5)	77.40 (32.8)
Laboratory tests	5.58 (8.9)	19.89 (8.4)
Echography	0.14 (0.2)	0.26 (0.1)
Sterilization	3.65 (5.8)	0.73 (0.3)
Food	14.33 (22.9)	4.63 (2.0)
TOTAL	62.58	236.05

**Table 7 ijerph-17-08085-t007:** Fixed Cost (US$).

Fixed Cost (US$)	NGO Hospital		Public Hospital	
	Per Year	Share for CS Per Month	Per Year	Share for CS Per Month
Depreciation buildings	151,371.16	986.14	704.48	26.58
Depreciation Equipment	47,191.03	630.70	16,046.67	456.26
Overhead staff costs	372,878.64	1180.48	255,350.03	2236.13
Other overhead costs	576,947.51	918.04	233,576.99	11,559.12
TOTAL	1,148,388.34	6444.58	505,678.16	8529.59

**Table 8 ijerph-17-08085-t008:** Average cost per CS (US$).

Average Cost Per CS	NGO Hospital			Public Hospital		
Number of patients	18	54	100	18	54	100
Variable costs	62.58	62.58	62.58	236.05	236.05	236.05
Fixed costs	358.03	119.34	64.45	473.87	157.96	85.30
Step fixed costs	262.61	184.09	170.94	145.75	78.13	76.26
Total	683.23	366.01	297.96	855.67	470.03	397.60

**Table 9 ijerph-17-08085-t009:** Sensitivity analysis.

Varied Parameter	Value of Parameter	Cost per CS NGO Hospital (US$)	Cost per CS Public Hospital (US$)
Basic		683.23	472.13
Variable cost	+10%/−10%	689.48/676.97	495.74/448.53
	+20%/−20%	695.74/670.71	519.35/424.92
Fixed cost	+10%/−10%	719.03/647.42	487.94/456.34
	+20%/−20%	754.83/611.62	503.73/440.55
Time for other delivery	0.5∙CS	690.04	473.46
	1.5∙CS	670.87	475.61
	2.0∙CS	666.01	473.93
Time for other duties	−10%	700.10	476.96
(% of duty time)	+10%	721.42	472.07
	+20%	704.65	474.72
Unproductive time	−5% DT; −10% OCT	680.57	474.24
	+10% DT; +10% OCT	719.35	484.09
	+20% DT; +20% OCT	736.35	496.84

DT = duty time; OCT = on-call time.

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
