# Peer review of "Costing of Cesarean Sections in a Government and a Non-Governmental Hospital in Cambodia—A Prerequisite for Efficient and Fair Comprehensive Obstetric Care"

_ijerph, 2020, doi:10.3390/ijerph17218085_

Round 1
Reviewer 1 Report
The paper is really well written and informative. I only have minor comments:
Introduction:
a. I would suggest that the authors more clearly differentiate review of reforms on service provision and on health financing side. Sentences are used sequentially to one another, which reflect reforms of different nature. It seems to me that Cambodia worked both to improve service delivery and financial accessibility. This dualism could be made more explicit.
b. There are few costing studies of health services, and CS in particular, in LMICs, so the value of the paper reaches beyond the Cambodian context alone. This point could be reinforced as well.
Methods:
The methodology is faultless and extremely rigorous. I particularly liked the fact that authors explicitly factored in the influence of frequency of CS in their analysis. I also really appreciate the data collection approach, based on direct observations and consultation of hospital records. This ensures high accuracy of the ABC approach.
I would just like to see that the authors appraise the limitations related to having worked exclusively in two hospitals. More specifically, how do we know that these hospitals are representative of the Cambodian context? How were the two hospitals selected and by whom?
Results:
The results section is extremely comprehensive and clear. No question on the tables being shown. They are self-explanatory and allow the reader to follow the story with ease. I appreciate the inclusion of the sensitivity analysis.
Discussion:
The findings are appraised adequately in relation to contextual elements, but I do feel that the authors could have tried to draw links to existing literature as well. For instance, to what extent are the patterns observed in this study similar to what observed in other context? There is a simple statement to this regard in the first paragraph of the discussion, but none as the single elements of the results are discussed in the second to six paragraph. If no appraisal in relation to literature is possible at all, I would suggest making it clear and justifying why. Appraisal in relation to literature only emerges again in the seventh paragraph of the discussion, which appears unusual and not fully credible. The evidence from other studies appearing in eight to tenth paragraphs could be integrated in the earlier sections of the discussion, to make the reading more fluent and not give the false impression that non appraisal in relation to literature is taking place.
A specific comment: authors claim that CS state may increase if quality of maternity services is focused. I do not understand this statement. Research indicates that increasing CS rate is subject to improving financial accessibility, by not only removing fees at point of care, but also caring for transport and time costs and improving referral systems. While I understand that increasing CS rate is only discussed in relation to its cost implications, the statements made to this regard should adequately reflect available literature.
Reviewer 2 Report
The authors are to be commended for a through description and analysis comparing CS costs between an NGO and a Public hospital. In effect you are comparing “apples” and “oranges” which is more than a sufficient rationale for the study.
The Introduction is clear and concise as to its purpose – determining the cost of CS using an activity-based approach. It is an inherently time-consuming and costly approach which explains the high frequency of top-down costing.
The Material and Methods section provides a considerable of explicit detail. While this study is submitted to the Health Economics section, the detail may be of major interest to economists but for administrators, government policy makers and other general audience readers it may be to dense. My preference is for the authors consider a more succinct presentation, especially for sections for 2.3, 2.4 and 2.5. Perhaps an appendix would be more appropriate for detailed presentation of text and charts/figures. Is Figure 1 necessary? Is the formula line 196 necessary. Alternately, the authors could make the detailed information available for those who are interested.
The Results are presented in a logical manner and provide sufficient detail. It is obvious that significant differences exist between the two hospitals. It is not so obvious as to why the differences exist given the context in which care is sought, differences in funding sources, possible differences as to why some patients seek an NGO and others seek a public hospital, etc. The authors offer possible reasons which would lead one to conduct further studies whether an activity-based or a more traditional top-down approach is used. Would the authors care to comment on any possible similarities in results for the two hospitals if both approaches were utilized?
To me ultimately the study is more about methodology than the reliability and validity of the results which is a significant achievement. The relatively small number of observations gives me some concern but in the present context it is not a major issue to me.
The authors are to be commended for a through description and analysis comparing CS costs between an NGO and a Public hospital. In effect you are comparing “apples” and “oranges” which is more than a sufficient rationale for the study.
The Introduction is clear and concise as to its purpose – determining the cost of CS using an activity-based approach. It is an inherently time-consuming and costly approach which explains the high frequency of top-down costing.
The Material and Methods section provides a considerable of explicit detail. While this study is submitted to the Health Economics section, the detail may be of major interest to economists but for administrators, government policy makers and other general audience readers it may be to dense. My preference is for the authors consider a more succinct presentation, especially for sections for 2.3, 2.4 and 2.5. Perhaps an appendix would be more appropriate for detailed presentation of text and charts/figures. Is Figure 1 necessary? Is the formula line 196 necessary. Alternately, the authors could make the detailed information available for those who are interested.
The Results are presented in a logical manner and provide sufficient detail. It is obvious that significant differences exist between the two hospitals. It is not so obvious as to why the differences exist given the context in which care is sought, differences in funding sources, possible differences as to why some patients seek an NGO and others seek a public hospital, etc. The authors offer possible reasons which would lead one to conduct further studies whether an activity-based or a more traditional top-down approach is used. Would the authors care to comment on any possible similarities in results for the two hospitals if both approaches were utilized?
To me ultimately the study is more about methodology than the reliability and validity of the results which is a significant achievement. The relatively small number of observations gives me some concern but in the present context it is not a major issue to me.
Author Response
please see the attachment (the same as for reviewer 1)
